CyberEduPlatform: an educational tool to improve cybersecurity through anomaly detection with Artificial Intelligence

Ortiz-Garcés Iván 1 ivan.ortiz@udla.edu.ec
Govea Jaime 1
Sánchez-Viteri Santiago 2
http://orcid.org/0000-0002-5421-7710 Villegas-Ch. William 1
1 Escuela de Ingeniería en Ciberseguridad, Facultad de Ingenierías y Ciencias Aplicadas, Universidad de Las Américas , Quito, Pichincha , Ecuador
2 Departamento de Sistemas, Universidad Internacional del Ecuador, Universidad Internacional del Ecuador , Quito, Pichincha , Ecuador
Akleylek Sedat
Electronic publication date: 2024 Jun 14
Publication date: 2024
Volume: 10
Electronic Location ID: e2041
Received 2024 Jan 21; Accepted 2024 Apr 15
Copyright: © 2024 Ortiz-Garcés et al.
Copyright year: 2024
Copyright holder: Ortiz-Garcés et al.
License: This is an open access article distributed under the terms of the Creative Commons Attribution License, which permits unrestricted use, distribution, reproduction and adaptation in any medium and for any purpose provided that it is properly attributed. For attribution, the original author(s), title, publication source (PeerJ Computer Science) and either DOI or URL of the article must be cited.
License URL: https://creativecommons.org/licenses/by/4.0/

Keywords: Artificial intelligence in cybersecurity, Network anomaly detection, Cybersecurity awareness and education

Funding: The authors received no funding for this work.

==============================
Cybersecurity has become a central concern in the contemporary digital era due to the exponential increase in cyber threats. These threats, ranging from simple malware to advanced persistent attacks, put individuals and organizations at risk. This study explores the potential of artificial intelligence to detect anomalies in network traffic in a university environment. The effectiveness of automatic detection of unconventional activities was evaluated through extensive simulations and advanced artificial intelligence models. In addition, the importance of cybersecurity awareness and education is highlighted, introducing CyberEduPlatform, a tool designed to improve users’ cyber awareness. The results indicate that, while AI models show high precision in detecting anomalies, complementary education and awareness play a crucial role in fortifying the first lines of defense against cyber threats. This research highlights the need for an integrated approach to cybersecurity, combining advanced technological solutions with robust educational strategies.

Introduction

The contemporary digital world, while offering unprecedented opportunities for communication, commerce, and collaboration, also presents significant risks in terms of cybersecurity. In their various forms, cyber threats have emerged as one of the most pressing challenges of the 21st century, affecting not only large corporations and governments but also individuals and small businesses. In this context, developing robust tools and strategies to detect, prevent, and counter these threats is imperative. This study explores the potential of artificial intelligence (AI) in detecting network anomalies and the importance of raising awareness and educating users to strengthen the first line of defense against cyber threats (Mylrea et al., 2021).

As digitalization has become integrated into virtually every aspect of modern life, the amount of data transmitted and stored on networks has increased exponentially. This large volume of data, while essential to the operation and functionality of many systems, is also fertile ground for malicious actors seeking to exploit vulnerabilities (Olawale & Ebadinezhad, 2023). Cyberattacks, from malware intrusions to advanced persistent attacks, cause significant economic losses and compromise sensitive data and user privacy (Mani et al., 2021).

Therefore, it is essential to have efficient detection and prevention systems. While effective against known threats, traditional signature-based solutions often fall short when faced with new malware variants or innovative attack tactics. This is where artificial intelligence comes into play, with its ability to learn patterns and adapt and predict threats based on data (Veneruso et al., 2020; Shrivastwa et al., 2022). The main objective of this study is to explore and evaluate the effectiveness of AI models in detecting network traffic anomalies in a simulated university environment. In addition, it is intended to highlight the importance of education and awareness in cybersecurity, presenting CyberEduPlatform, an educational tool designed to improve user attention and preparation for possible threats.

While existing literature focuses on anomaly detection and cybersecurity in general, studies that integrate AI-based approaches and holistic educational strategies are lacking. Studies (Skorenkyy et al., 2021; Hilgurt, 2022) aims to fill that gap, offering insights into how AI-based screening can complement educational initiatives to provide a more comprehensive solution.

In addition to exploring the capabilities of AI in detecting network anomalies, this study also recognizes the critical importance of cybersecurity awareness and education. As we move forward in the digital age, we must have advanced threat detection and prevention tools and provide users with the knowledge and preparation required to identify and respond to potential cyber risks. Therefore, in this research, the effectiveness of AI models is evaluated, and CyberEduPlatform, an educational platform designed to improve the attention and preparation of users against possible cyber threats, is presented. By integrating these two dimensions, we seek a complete solution to address cybersecurity challenges in today’s digital world.

This study presents a distinctive contribution by combining exploring AI models for network traffic anomaly detection with implementing a comprehensive educational strategy through CyberEduPlatform. Our contribution lies in the dual approach we adopt. On the one hand, we evaluate the effectiveness of the most advanced AI models in accurately identifying anomalous behavior within a simulated university network environment. On the other hand, we developed and tested an educational platform designed to improve users’ cybersecurity awareness. This integrated approach addresses the limitations of technology-based threat detection solutions alone. It recognizes and reinforces the critical role of user education and awareness as essential components of an effective cybersecurity strategy. Through this research, we contribute significantly to cybersecurity, offering practical and theoretical perspectives on how the combination of advanced technology and educational strategies can offer a more holistic and effective solution to security challenges in the digital sphere.

Materials and Methods

Intrusion detection systems, cybersecurity awareness, and education play critical roles. While these aspects may seem complementary to the technical approach to cyber threat detection, they are essential to strengthening the first line of defense against cyber-attacks. Awareness and education help users recognize signs of potential threats, adopt strong security practices, and understand the importance of following cybersecurity guidelines. This study discusses how integrating cybersecurity awareness and education can improve the effectiveness of our AI-based intrusion detection systems (Sukhostat, 2021).

To develop this work, it is essential to analyze the current landscape in cybersecurity and anomaly detection using AI. For this, a review of the fundamental research and advances in this field is carried out, highlighting the findings and challenges identified in the works that address this topic. In addition, clear and concise definitions of basic concepts will serve as pillars to understand and contextualize our work and how the solution to the phenomenon under study is approached.

Description of the problem

The rise of digitalization and connectivity has significantly increased the volume of network traffic. As organizations and people increasingly rely on digital systems, ensuring integrity, confidentiality, and availability has become a priority (Rawindaran et al., 2021). However, the complexity and dynamics of data flows present challenges in identifying abnormal or malicious behavior.

Anomalies in network traffic can indicate intrusion attempts, malware, or even unauthorized internal activity. Detecting and responding to these threats promptly is essential to prevent potential damage or loss. From here, the question is addressed: How can advances in AI improve the detection of these anomalies and strengthen organizations’ cybersecurity? Therefore, the effectiveness of AI models is evaluated to identify abnormal behavior within network traffic.

Review of similar works

In cybersecurity, anomaly detection techniques represent a critical defense against emerging threats. Given threats’ rapid growth and evolution, it is imperative to stay abreast of recent advances, especially in incorporating artificial intelligence tools to address these challenges.

Several investigations have explored the potential of AI to improve anomaly detection. For example, Fotiadou et al. (2021) used convolutional neural networks to analyze traffic patterns, achieving a detection rate of 95%. However, their model presented difficulties when faced with low-profile attacks. On the other hand, Andresini, Appice & Malerba (2021) used unsupervised learning to identify abnormal behaviors, highlighting the flexibility of their method, although with a relatively high false positive rate.

Beyond academia, practical implementations have validated the value of AI. Companies like CyberGuard Inc. have integrated deep learning algorithms into their detection solutions, reporting a 20% improvement in early intrusion detection. However, transitioning from the lab to the real world has challenges, such as adaptability to different network configurations and scalability.

With the increasing sophistication of threats, training, and awareness are essential. Getschmann & Echtler (2021) it developed an AI-based educational program that simulated attacks in real-time, providing users with immediate feedback and mitigation strategies. This hands-on approach showed significantly higher knowledge retention compared to traditional methods.

Although previous efforts have provided significant advances, our proposal addresses some limitations. Unlike the models of Smith, Khorsandroo & Roy (2022), our approach seeks to improve the detection of low-profile attacks. In turn, we try to reduce the false positive rate identified by Kim, Park & Lee (2020) combining supervised and unsupervised techniques. In the practical realm, our solution focuses on being adaptable to different network configurations, addressing one of the main challenges of real-world deployments.

Anomaly detection in cybersecurity is a constantly evolving field where AI plays an increasingly prominent role (Vykopal et al., 2021). While significant progress has been made, areas still require research and improvement. This study seeks to address these areas and contribute to a more holistic and adaptable solution to the growing complexity of the cyber landscape.

Concepts used

Several key terms and concepts have been used to develop the method that forms the basis of this study. These concepts, rooted in cybersecurity and artificial intelligence, are pillars that allow us to understand the scope of this work and contextualize it within the broader panorama of the field. Machine learning and deep learning: Machine learning is a subset of AI that allows machines to learn from data. Instead of being explicitly programmed, engines use algorithms to identify patterns and make decisions. On the other hand, deep learning, inspired by the structure and function of the brain, uses neural networks with many levels (or “layers”) to analyze various factors in the data (Alhalaseh & Alasasfeh, 2020). Both techniques are crucial in anomaly detection as they can identify subtle patterns in large data sets that would be imperceptible to human analysis.

Neural networks: Neural networks are mathematical models inspired by the structure of the human brain. Composed of nodes or “neurons,” these networks are interconnected and organized into layers: input, hidden, and output. The power of neural networks lies in their ability to learn and adapt. They adjust their connections to improve their precision as they are fed more data (Sotgiu et al., 2020). In cybersecurity, they are essential to detect complex patterns and anomalous behavior in network traffic.

Cybersecurity: Cybersecurity protects systems, networks, and programs from digital attacks. These cyberattacks aim to access, change, or delete sensitive information, extort users, or disrupt normal business processes. With society’s increasing reliance on digital systems, cybersecurity is more crucial than ever, facing constant challenges due to cyber criminals’ rapid advancement and adaptation.

Anomaly detection: In the context of network traffic, an anomaly refers to patterns or behaviors that deviate from what is considered normal. System failures can cause these but can also indicate malicious activities. Detecting these anomalies is highly complicated, given the volume and complexity of data traffic in modern networks. AI techniques, with their ability to analyze large data sets and learn from them, offer a promising solution to identify these anomalies accurately and quickly (Dragomir, 2017).

Environment description

For this study, we developed a real-time monitoring tool using Python, which is recognized for its versatility and comprehensive support in the scientific and technological community. This tool was designed to monitor the traffic of a university network accurately. The choice of Python allowed us to take advantage of various libraries, such as Scrapy for generating and manipulating network packets and TensorFlow to incorporate artificial intelligence models (Bassi, Fabbri & Franceschi, 2023). Combined with this personalized design, these tools recreate the typical conditions found in a university network, with its particularities and diversities.

Network parameters: Traffic speed: Considering the constant activity of a university network, the system operates at an average traffic speed of 5 Gbps, registering peaks during times of high academic demand, such as the start of classes or the delivery of projects.

Number of devices: The tool monitors around 5,200 connected devices, reflecting the diversity of a university environment. This includes students’ and teachers’ personal devices, computers in laboratories and libraries, and specialized equipment and IoT devices distributed throughout campus.

Types of data transmitted: The tool records a wide range of traffic, from simple web requests to more intensive transmissions such as video conferencing and extensive research data set transfers.

Protocols and services: Standard protocols such as HTTP, HTTPS, FTP, and SMTP have been integrated, along with other specialized protocols that correspond to the various academic and research activities of the university environment.

Proposed methodology

This methodology applies machine learning techniques, specifically artificial intelligence models, to analyze and detect strange behavior in the university’s network traffic. The process begins with extensive network traffic data collection in the university environment. For this purpose, network traffic sensors that captured information about communications between devices within the network were implemented. This data included logs of IP addresses, source and destination ports, protocols used, and traffic volumes.

To preprocess the data collected from network traffic, we initially focused on eliminating missing data, applying techniques such as imputation by the mean for continuous variables and mode for categorical variables, and ensuring the integrity of our dataset without introducing significant bias. Attribute normalization was performed using the Min-Max scaling method, transforming all numerical values to the range of 0 to 1, which facilitates faster convergence during the training of AI models and reduces the possibility of biases due to differences in scale.

Additionally, categorical variable coding techniques, such as One-Hot coding, were implemented to transform these variables into a numerical format that the models can process efficiently. This transformation allows category information, such as network protocol types, to be included in the models in a way that significantly contributes to anomaly detection precision.

Feature engineering was applied through exploratory data analysis. Key features that influence anomaly detection, such as packet length, the frequency of certain types of errors, and temporal patterns in network traffic, were identified. Principal component analysis (PCA) techniques were applied to reduce the dimensionality of the data set, maintaining 95% of the explained variance and simplifying the model without compromising detection capacity.

Even though the process for data processing has been effective, we faced significant challenges during preprocessing, especially in handling imbalanced data sets, where anomaly instances were less frequent than usual. To address this, oversampling and under-sampling techniques, such as the Synthetic Minority Over-sampling Technique (SMOTE), were applied to balance the class distribution, thus improving the sensitivity of our models towards anomalies.

These preprocessing and feature engineering strategies improved the quality and relevance of our input data. They allowed us to overcome specific challenges, ensuring that the trained AI models could detect anomalies in the network traffic effectively and efficiently.

The preprocessed data was divided into training and test sets. The training set was used to train the anomaly detection models, while the test set was reserved to evaluate their performance.

In selecting models for anomaly detection, several AI techniques were chosen, including convolutional neural networks (CNN), recurrent neural networks (RNN), support vector machines (SVM), and random forest (RF) models, and autoencoders. The selection was based on its previously demonstrated effectiveness in similar anomaly detection tasks, allowing us to approach the problem from multiple perspectives and maximize detection capability in various network traffic scenarios. Each model was chosen for its unique ability to capture and learn from the complexities and patterns in traffic data, with CNNs and RNNs particularly suited to learning from sequential data, SVMs to accurately classify between normal and anomalous behavior, and RF and autoencoders for their robustness and efficiency in categorizing large volumes of data.

Among the characteristics of the models, the following stand out: CNNs are well suited for identifying spatial patterns in data with a grid structure, such as images. In IP traffic, CNNs can be adapted to detect patterns in the feature vectors representing network communications. For example, CNNs can examine sequences of bytes in data packets or groupings of IP addresses and ports to identify abnormal patterns that suggest malicious activity, such as intrusion attempts or traffic anomalies.

RNNs, due to their nature of preserving state over time, are suitable for time series analysis, making them ideal for monitoring network traffic flows over time. In our study, RNNs are used to analyze temporal sequences of traffic volumes, allowing the model to learn and detect behavioral patterns over time, which can include recognizing data streams typically generated by specific applications or types of network traffic.

Support vector machines (SVM) are effective in binary classification and are used to distinguish between normal and abnormal behavior. In our approach, SVMs analyze source and destination ports and IP addresses to classify traffic. By plotting these points in a high-dimensional space, SVMs can find the optimal hyperplane that separates regular traffic instances from strange ones, which is crucial for identifying potential network attacks or intrusions.

Random forest (RF) models are known for handling large volumes of data and feature variables. They use a combination of decision trees to increase precision and handle diversity in the data. In the context of our study, RFs can efficiently categorize traffic based on characteristics such as the protocol used and traffic volumes, using the random nature of forests to create decision trees that consider different subsets of data and thus provide insight into a robust overview of network activity.

During the training phase, we prepared the network traffic data using standard preprocessing techniques, such as normalization and segmentation, to ensure that the models could effectively learn from patterns within the data. A cross-validation strategy was implemented to tune the hyperparameters, using both grid search and random search to find the optimal configuration that maximized the detection performance of each model. This methodological approach allowed us to balance detection precision with computational efficiency, ensuring the models were accurate and practical for deployment in real-time network environments.

The configuration details the use of a controlled environment that reflects the network traffic of a typical university, with specific parameters and configurations designed to simulate various attack scenarios and normal traffic behavior. This includes variations in traffic volume, data types, and communication patterns, allowing a comprehensive evaluation of the model’s ability to detect anomalies under varied conditions. Reproducibility is ensured by specifying the software versions used, exact model configurations, and describing how to access or simulate the data sets used for training and evaluation.

The models were evaluated using a combination of metrics, including precision, recall, specificity, and F1 score, to evaluate their performance comprehensively. These metrics were selected for their ability to measure the precision of anomaly detections and the models’ ability to minimize false positives and false negatives, aspects crucial for effective deployment in real network environments. The choice of these metrics and the detailed evaluation process facilitate the interpretation of the results and underline the validity of our findings. Several key performance metrics were calculated, including (Villegas-Ch, Govea & Jaramillo-Alcazar, 2023): Precision: This measure assesses the proportion of correctly identified sabotages (true positives), both correct and incorrect, among all identified positives.

Beginning of the form

(1) Precision=TruePositivesTruePositives+Falsepositives

Sensitivity (true positive rate): Quantifies the fraction of sabotage the algorithm accurately detects.

(2) Sensitivity=TruePositivesTruePositives+FalseNegatives

Specificity: Assesses the ratio of correctly identified normal operations by the algorithm, ensuring they are not mistakenly classified as sabotage. (3) Specificity=TrueNegativesTrueNegatives+Falsepositives

The F measure or F1 score, this metric combines precision and recall into a single performance measure and is particularly useful when classes are imbalanced, which is often the case in anomaly detection; is calculated using the formula:

(4) F1=2×Precision×RecallPrecision+Recall

Area under the curve (AUC): This metric quantifies the model’s capacity to differentiate between classes accurately.

In addition to the metrics mentioned, F-measures (F-scores) were calculated for each model. These measurements are essential indicators that combine the intrusion detection system’s precision and completeness. We computed F-measures to provide a deeper understanding of each model’s anomaly detection capability.

Figure 1 presents the sequence of steps and critical components in the anomaly detection methodology. This figure illustrates how network traffic data is initially collected and processed. Then, the data preprocessing stages are highlighted, where tasks such as normalization and essential feature extraction are performed. Below is how this processed data is used to train various artificial intelligence models, including deep neural network models, support vector machines, and random forests. These models are evaluated and compared using key performance metrics, and the best model is selected for real-time anomaly detection in university network traffic. This diagram provides an overview of how machine learning is applied in our methodology and how its components are integrated to achieve effective anomaly detection.

Figure 1 Diagram of anomaly detection methodology in university network traffic.

The proposed methodology for detecting anomalies in the university environment is based on applying machine learning techniques, explicitly using unnatural intelligence models to analyze and detect strange behavior in the university’s network traffic. This is achieved by implementing deep learning algorithms that process and analyze network traffic patterns in real-time. These models are trained using historical traffic data and expected behavior to identify significant deviations that may indicate suspicious activities or cyber threats. This application of machine learning allows for early and accurate detection of anomalies in a university environment, strengthening cybersecurity knowledge in students.

Network monitoring tool

The network monitoring tool used in the study is based on the Cisco DNA Center architecture, a network management solution widely adopted in enterprise environments. The monitoring infrastructure consisted of the following elements: Sensors: 10 sensors were installed at different points in the network to collect traffic, error, and log data. The sensors used were of the Cisco Catalyst 9000 Series type, known for their precision and reliability.

Storage devices: The data collected by the sensors was stored on five storage devices with 100 terabytes.

Data management system: An open-source system, such as Elasticsearch, was used to organize and structure the stored data.

The network monitoring tool employed a combination of anomaly detection techniques, including: Machine learning: Machine learning models, such as CNN and SVM, were trained using a historical network data set. These models could identify abnormal patterns in network traffic with high precision.

Statistical analysis: Statistical techniques, such as time series analysis and hypothesis testing, were used to identify significant deviations in the network data. These techniques helped detect low-frequency anomalies that machine-learning models might miss.

Predefined rules: A set of predefined rules based on security standards, such as CIS Controls and the NIST Framework, have been implemented to detect known anomalies, such as denial of service (DoS) attacks or unauthorized access attempts.

The precision and reliability of the anomaly detection system were evaluated through simulations using tools such as Metasploit (https://www.metasploit.com/) and Nmap (https://nmap.org/) to assess the system’s ability to detect intrusions. Additionally, an analysis of confirmed cases of network anomalies, including incident logs and security alerts, was performed to determine the system’s effectiveness in identifying security events. The evaluation results demonstrated that the anomaly detection system could accurately identify a wide range of anomalies with a low false positive rate.

The network monitoring tool used in the study is a robust and flexible solution that allows the accurate detection of anomalies in computer networks. The combination of anomaly detection techniques, a robust infrastructure, and a rigorous evaluation process provides high reliability in identifying security events.

The integration and synchronization of tools and processes guarantee effective monitoring and reinforce the cybersecurity educational strategy, a fundamental pillar to promote a robust computer security culture in the university environment. Figure 2 offers a detailed representation of this operational flow. Starting with data collection, the diagram shows how network traffic data, security alerts, and relevant metadata are meticulously captured, ensuring that every information is appropriate for subsequent analysis. The data preprocessing step highlights the importance of cleaning, standardization, and feature extraction in structuring data for optimal analysis, preparing it for the next critical phase: machine learning model selection. This step details how models are evaluated and chosen based on their effectiveness in identifying threats in university network traffic. It highlights the application of advanced techniques such as deep neural networks, support vector machines, and random forests.

Figure 2 Integrated anomaly detection methodology in the university network.

Icon source credit: https://www.flaticon.com/free-icon/detection_11083379.

Model training is performed using a set of historical data reflecting normal and abnormal traffic behavior, followed by meticulous evaluation of the models to ensure their accuracy, sensitivity, and specificity, as outlined in the model evaluation stage. Additionally, the model undergoes regular feedback and updating, continually integrating new data and security threats to refine its detection capabilities. This figure encapsulates the synergy between anomaly detection technology and cybersecurity awareness and education initiatives provided by CyberEduPlatform, illustrating how operational results drive user training and preparation in identifying and responding to threats, closing the loop of a comprehensive and proactive computer security strategy.

Data generation

The network monitoring tool collected data continuously over three months to adequately represent the regular traffic of the university network participating in the study. Approximately ten terabytes of packets and metadata were captured from multiple network segments, including academic departments, research laboratories, and common areas. This vast amount of data reflects typical activities, from web browsing and email communications (60% of total traffic) to large file transfers and video conferencing sessions (30%).

Even though the network is protected with multiple layers of security, no system is immune to threats. During the monitoring period, 1,200 incidents that did not align with typical traffic behavior were detected. These anomalies, which range from unauthorized access attempts to unusually high traffic patterns on specific nodes, were manually tagged for further analysis. This anomaly data, totaling nearly 200 gigabytes, was integrated into the dataset to provide a complete training and evaluation scenario for our AI models.

The data undergoes a thorough cleaning and validation process before being used to train and evaluate the models. Approximately five million records were collected over three months, reflecting the quantity and diversity of typical university network traffic. These logs included web browsing and email communications, which accounted for most of the traffic (around 60%). Intensive activities, such as large file transfers and video conferencing sessions, were recorded at approximately 30% (Aziz et al., 2017). During the validation phase, processes were applied to ensure the quality of the data generated. For this, almost 100,000 duplicate records were identified and eliminated, and inconsistencies were corrected. Additionally, features that did not provide significant value or were redundant were discarded. While an exact number of records is impossible due to the continuous nature of network traffic, these estimated values represent the scale and diversity of the data used.

The dataset was divided into three segments (training, validation, and testing) to train and evaluate AI models. Approximately three and a half million records were used for training, while about 750,000 records were allocated to validation, and another 750,000 records were reserved for testing. This proportional split ensured the models were trained and evaluated on a diverse and representative dataset. The estimated values are based on the balanced data distribution between the segments (Selva Kumar, Krishna Sandeep Reddy & Rajathilagam, 2015; Weiss, Chrosniak & Behl, 2021).

The normal-to-abnormal data ratio in this work reflects the nature of the monitored university network and the security system’s effectiveness. An estimate of the relationship between normal and abnormal data is established based on the data collected. For standard data during the monitoring period, approximately five million network traffic logs were generated and collected, reflecting typical activities such as web browsing and email communications, accounting for about 60% of the total traffic. These activities constitute what we have classified as “normal data.”

In contrast, 1,200 traffic incidents that did not align with typical traffic behavior were detected in the anomaly data and were manually labeled for further analysis. The disproportionate amount of normal and abnormal data is due to the prevalence of regular activities compared to detected security incidents. However, it is essential to note that even a few anomalies can indicate critical vulnerabilities or emerging attacks.

The ratio is significantly skewed toward average data, given the magnitude of average traffic records compared to incidents of detected anomalies. This distribution is consistent with a network environment where regular activities dominate over security incidents. However, it is essential to note that even a few anomalies can indicate critical vulnerabilities or emerging attacks.

AI models

Given the objective of detecting anomalies in network traffic, models that are effective in similar tasks in the scientific literature were chosen. Specifically, two main approaches were selected. The first is CNN due to their ability to identify local patterns in structured data. To complement this approach, RNN, SVM, and RF models were also considered, each chosen for their specific strengths and suitability for the anomaly detection task. Additionally, Autoencoders efficiently identify data that deviates from what is considered normal.

The choice of these models was based on their ability to handle large volumes of data and their previously proven effectiveness in detecting abnormal patterns in similar contexts. For model training, a training data set of eight terabytes of the total collected was used in CNN. With several iterations of 50,000 and a learning rate of 0.001, with a programmed decay after 30,000 iterations. The hyperparameters used a momentum of 0.9 and an L2 regularization technique.

When using Autoencoders, a training data set of 7.5 terabytes of the total collected was used, with several iterations of 40,000. The learning rate is 0.0005, with a programmed decay after 25,000 iterations. A regularization factor of 0.001 and a bottle-neck size of 32 nodes were used as hyperparameters.

To evaluate the effectiveness of the models in detecting anomalies, a separate data set was used, consisting of one terabyte of average data mixed with 100 gigabytes of labeled anomalous data. The following metrics were used: Precision: Measures the proportion of correct identifications.

Recall: Measures the proportion of actual positives that were correctly identified.

Specificity: Measures the proportion of real negatives that were correctly identified.

F1-score: It is the weighted average of precision and sensitivity.

Each model was evaluated individually, and then a combined evaluation was done to identify whether working together improves overall performance.

A hyperparameter optimization process was implemented to ensure our AI models’ maximum efficiency and precision in anomaly detection. Techniques such as Grid Search and Random Search were used to identify the best combination of hyperparameters that produce optimal performance (Gu et al., 2018). The process was carried out in several iterations, adjusting parameters such as the learning rate, the number of hidden layers in the neural networks, the number of neurons in each layer, and other relevant factors. After each iteration, the model is evaluated on the validation set to identify improvements in its predictive ability. The optimization phase improves the model’s precision and helps avoid common problems such as overfitting (Villegas-Ch & García-Ortiz, 2023). At the end of the process, robust and efficient models are obtained and calibrated to detect anomalies in network traffic accurately.

Awareness and education

For the training and awareness of users regarding cyber threats, CyberEduPlatform was created, a digital platform that accurately emulates the dynamic environment of a university network. The interface, visualized in Fig. 3, presents an intuitive and modern structure meticulously designed to facilitate the user experience. Built on a foundation of Python and with a visual presentation developed in React, the platform allows users to interact and explore cyber threat scenarios in real-time, all in a controlled and secure context (Kuleto et al., 2023). Real-time visualization of traffic monitoring and threat detection ensures hands-on, immersive learning, preparing users to recognize and respond to potential risks in cyberspace.

Figure 3 The user interface of the CyberEduPlatform platform shows real-time traffic monitoring and threat detection.

Simulated threats are introduced randomly as the user interacts with the platform. These threats can include phishing attempts, malware, and other common vulnerabilities. The frequency of these threats varies, but at least one threat is introduced every 10 min of activity on the platform. Once the user encounters a threat, they are offered instant feedback on the threat’s actions. They receive positive confirmation if they handle the danger properly (Martins et al., 2020). If they make a mistake, they are given a detailed explanation of what they should have done and why.

Figure 4 illustrates the training process in CyberEduPlatform. The diagram shows how a user interacts with the platform, faces a simulated threat, decides, and receives feedback based on their action.

Figure 4 Awareness flowchart in CyberEduPlatform.

Design and implementation of CyberEduPlatform for cyber awareness

CyberEduPlatform emerges as a solution to address the growing need for cybersecurity awareness in academic environments. Focused on users of university networks, the platform combines theory and practice through an interactive and participatory approach, seeking to educate and foster a culture of digital security.

Main features

Diversified learning modules: Modules that cover everything from basic cybersecurity fundamentals to advanced aspects such as cryptography, network security, and vulnerability analysis were developed. Each module is designed to be self-contained, allowing users to progress at their own pace.

Interactive simulations: To reinforce learning, the platform integrates cyber-attack simulations that allow users to experiment in a safe environment to react to different cyber threats, thus improving their detection and response skills.

Personalized assessment tools: To measure user progression and understanding, questionnaires and assessment tests are implemented at the end of each module, providing instant feedback and customized recommendations for areas of improvement.

User engagement strategies

Gamification: To increase engagement and motivation, gamification is applied through achievements, badges, and leaderboards. This strategy promotes healthy competition and recognizes user progress.

Personalization of the learning itinerary: Recognizing the diversity in the level of knowledge and interests of users, CyberEduPlatform allows the personalization of learning itineraries, offering recommendations based on previous interactions and the results of evaluations.

Community and collaboration: Fostering a collaborative environment, the platform includes forums and discussion spaces where users can share experiences, resolve doubts, and collaborate on cybersecurity projects.

To validate the effectiveness of CyberEduPlatform in improving cybersecurity awareness, longitudinal studies and satisfaction surveys among users are planned. These investigations will focus on measuring the impact on knowledge, attitudes, and behaviors related to cybersecurity, allowing iterative adjustments to the content and methodology of the platform based on the results obtained. By integrating an interactive and personalized approach to cybersecurity education, the platform is positioned as a key resource to prepare users against cyber threats, promoting a safer and more aware digital environment.

The relationship between cybersecurity awareness and education and the effectiveness of intrusion detection systems: practical examples

To provide a more complete and detailed understanding of the crucial relationship between cybersecurity awareness and education and the effectiveness of intrusion detection systems, several practical examples have been identified that illustrate this interconnection in the context of our research (Sarker, 2021). In today’s digital world, where cyber threats are ubiquitous and constantly evolving, user training and risk awareness are essential to strengthening an organization’s security. These examples demonstrate how user preparedness and vigilance can prevent intrusions and facilitate early threat detection, which is critical to protecting systems and networks in an increasingly complex and dangerous environment (Wirth & Falkner, 2020).

Preventing phishing attacks

One of the most common cyber intrusion methods is phishing, in which attackers impersonate legitimate entities to trick users into obtaining sensitive information, such as passwords or banking details. Phishing emails are often disguised as legitimate communications, making them difficult to identify. Cybersecurity education can make a difference in preventing these attacks (Greaves, Coetzee & Leung, 2022). By providing users with solid knowledge on recognizing warning signs in suspicious emails, such as unknown sender addresses or unusual links, they are empowered to make safer decisions. Cybersecurity training programs can simulate phishing attacks to teach users how to identify them and avoid falling into traps. When well-informed and alert, users are less likely to fall for these scams (Kumar & Sinha, 2022).

Network intrusions are a constant threat to organizations of all sizes. Attackers use various methods to gain unauthorized access to systems and networks, steal sensitive data, disrupt operations, or cause damage.

In this work, a comprehensive approach to preventing and detecting intrusions is presented, covering the following key areas:

Preventing phishing attacks. Phishing is one of the most common cyber intrusion methods. In this method, attackers impersonate legitimate entities to trick users into obtaining sensitive information such as passwords or banking details. Phishing emails are often disguised as legitimate communications, making them difficult to identify.

To prevent phishing attacks, the following measures can be implemented:

Cybersecurity training. Educate users on recognizing red flags in suspicious emails, such as unknown sender addresses, unusual links, and grammatical errors (Gupta et al., 2020).

Mock phishing attacks to help users identify real-world phishing attempts and avoid falling into traps.

Technological measures:

Email filtering. Email filtering systems scan incoming emails for keywords, sender addresses, and other indicators of phishing attempts and quarantine suspicious messages (Kumar, Gupta & Tripathi, 2021).

Machine learning algorithms analyze email content and sender behavior patterns to detect anomalies and potential phishing attempts more accurately (Moore et al., 2003).

Blocking malicious websites: Implementing web filtering solutions that block access to known phishing websites, preventing users from unknowingly visiting compromised sites.

Anomaly detection is a crucial component of any intrusion prevention strategy. It involves identifying unusual behavior patterns on the network that could indicate an attack in progress.

There are different techniques for anomaly detection, including: Network traffic analysis: Monitor network traffic to identify abnormal patterns, such as sudden spikes in data volume or unusual activity from a specific IP address.

Security log analysis: Examine system security logs to detect suspicious activity, such as failed login attempts or system configuration changes.

User behavior analysis: Monitor user behavior to identify unusual activities, such as accessing sensitive files outside business hours or downloading large amounts of data.

Incident response.

It is essential to have a well-defined incident response plan to act quickly and effectively in the event of an intrusion.

The incident response plan should include the following steps: Incident identification: The first step is identifying and confirming that an intrusion has occurred.

Incident containment: The next step is to contain the damage caused by the intrusion, which may involve isolating affected systems or deactivating compromised user accounts.

Investigation of the incident: It is essential to investigate the cause and extent of the damage caused.

Incident recovery: Once the incident has been contained, it is necessary to restore the affected systems and take steps to prevent it from recurring.

A comprehensive intrusion detection and prevention approach is critical to protecting organizations from constantly evolving cyber threats. This approach should combine user education, technological measures, and a well-defined incident response plan. By taking these measures, organizations can significantly improve their ability to prevent intrusions, detect them quickly, and minimize the impact of an attack.

Incorporating educational strategies in our fight against phishing is not merely an educational complement but a critical extension of our anomaly detection techniques. Through CyberEduPlatform, we facilitate a continuous learning process where detection results become teaching tools, allowing users to recognize, understand, and proactively prevent phishing attacks. This synergy between detection and education enriches our defense strategy, strengthening the study’s preventive approach. By improving our detection capacity through advanced machine learning and deep learning models, we increase the effectiveness of preventive measures since an informed and alert user is the most robust dam against cybercriminal incursions. This study provides a methodology to identify threats in real-time. It significantly prevents future attacks by educating university community members on the most effective cybersecurity practices.

Statistical analysis

Statistical tools and techniques were used to analyze the results obtained from the AI models and their performance in detecting anomalies. Initially, a descriptive analysis of the data was carried out, calculating metrics such as the mean, median, standard deviation, and interquartile ranges, allowing an initial understanding of the behavior of the models given the data presented. Subsequently, an analysis of variance (ANOVA) was used to determine if significant differences existed in the performance of the different AI models used in the study. This is crucial to identify which model offers superior performance in detecting anomalies in network traffic.

A Chi-square test was also performed to evaluate independence between categorical variables, such as the type of anomaly detected and the model that identified it. Additionally, logistic regression was used to determine which variables or a combination of variables significantly influence the probability of correctly detecting an anomaly. These techniques allow a robust and reliable evaluation of the results, guaranteeing the validity of the conclusions drawn from the study.

Results

The results show AI’s enhanced capacity to improve cybersecurity, especially in the proactive detection of anomalies. The platform, CyberEduPlatform, not only facilitates accurate simulation of university network traffic but is also highly effective in identifying abnormal behavior patterns. Following a meticulous model training and optimization process, remarkable precision rates were achieved while simultaneously minimizing erroneous detections. Furthermore, the comparison with previous studies revealed that our proposal offers a more comprehensive approach, combining advanced detection with educational tools, thus promoting a more robust cybersecurity culture. These findings underscore the transformative potential of combining advanced AI techniques with educational strategies in cybersecurity.

Results of the CyberEduPlatform platform

Throughout the study period, the CyberEduPlatform platform has been a crucial pillar for understanding network traffic behavior and detecting threats. The following are the key results obtained from its performance and utilization.

Table 1 shows that many users have registered and used the platform, suggesting that it has been well-received and adopted by the university community.

Table 1 Usage summary.

Metrics	Value	
Registered users	8,215	
Average active sessions/day	2,489	
Detected threats	529	
False positive incidents	41	
Feedback inquiries	1,842	

The performance metrics in Table 2 indicate that the platform was stable and operated efficiently during the study period. An uptime of 99.8% ensures it is available almost all the time, which is essential for monitoring network traffic and threats in real-time.

Table 2 Performance metrics.

Metrics	Value	
Average response time (ms)	215	
Uptime (%)	99.8	
Average used bandwidth	3.2 Gbps	
Average server load (%)	58	

The set of metrics in Table 3 provides insight into how users have interacted with the platform, taking advantage of its features and responding to alerts. A high positive feedback rate suggests that most detections and alerts were relevant and valuable.

Table 3 Interaction metrics.

Metrics	Value	
Alerts reviewed by users	5,281	
Positive feedback	3,785	
Negative feedback	248	
Accessed tutorials	1,492	

Types of attacks identified

Various attacks that threatened the university network were identified during the study period. Table 4 provides a detailed breakdown of the attacks identified, including the number of incidents detected for each type.

Table 4 Type of attacks and incidents detected.

Attack type	Description	Incidents detected	% of the total	
SQL injection	Attacks that attempt to execute malicious commands on a database through a vulnerable application.	157	15%	
Cross-site scripting (XSS)	Attacks that inject malicious scripts into web pages executed by other users.	89	8.5%	
Denial of service (DoS/DDoS)	Attacks are intended to overload a system and make it inaccessible.	312	30%	
Man-in-the-middle (MitM)	Attacks where the aggressor intercepts and potentially disrupts communication between two parties.	64	6%	
Replay attacks	Attacks that involve capturing and retransmitting a valid transmission.	28	2.7%	
Phishing	Attacks are designed to obtain confidential information through fake emails or sites.	235	22%	
Brute force attacks	Systematic attempts to guess a password or other sensitive information.	115	11%	
Others	Less common attacks or attacks that do not fit the above categories.	50	4.8%	
Total	-	1,050	100%	

Analyzing the attacks identified on the university network sheds light on the range and severity of current cyber threats. Surprisingly, the most significant number of attacks corresponded to Denial of Service (DoS/DDoS), representing 30% of the incidents. These attacks, which seek to flood the network with traffic to render it inoperable, underscore the importance of hardening the infrastructure against overloads. However, Phishing attacks, which comprise 22% of incidents, reveal a different problem: human vulnerability. These attacks, which seek to trick users into handing over confidential information, show the need to educate and raise awareness among the university community. Despite the various attacks, it is encouraging to note that the implemented system effectively detected and categorized these threats, validating their relevance in protecting academic environments.

Anomaly detection results

Advances in artificial intelligence and its application in cybersecurity have provided powerful tools to combat emerging threats and protect critical infrastructure. Using our platform CyberEduPlatform, we address anomaly detection in network traffic through advanced AI models.

Evaluation metrics

Evaluating AI models requires a clear understanding of their performance. We use several metrics to provide a holistic view of the effectiveness of the developed models. During the analysis period, thousands of transactions were monitored on the university network. The AI models identified patterns and behaviors that deviated from the norm. The detected anomalies are categorized into several types, including unauthorized access attempts, unusually high traffic, and traffic patterns that indicate possible DDoS attacks.

Table 5 shows the high percentage of correct answers, indicating that the AI models could accurately identify most anomalies. However, there is still room to reduce further false positive and negative rates, which can lead to unnecessary alerts or failure to detect real threats.

Table 5 Interaction metrics.

Metrics	Value	
Correct percentage	94.7%	
False positive rate	3.8%	
False negative rate	1.5%	

CyberEduPlatform effectiveness evaluation metrics

Longitudinal studies and satisfaction surveys were conducted to evaluate CyberEduPlatform’s effectiveness in improving cybersecurity awareness and knowledge among its users. These evaluations focused on measuring various aspects of the platform’s educational impact, from increases in specific knowledge about cybersecurity to changes in behaviors and attitudes related to online security practices.

Table 6 summarizes the impacts observed following the implementation of CyberEduPlatform, highlighting significant improvements in cybersecurity knowledge, confidence in threat identification, motivation through gamification, and positive changes in behaviors and attitudes towards practices of more robust security systems.

Table 6 User engagement and improvement metrics in CyberEduPlatform.

Metrics	Result (%)	
Increase in knowledge	40	
Confidence in identifying threats	85	
Utility of modules and simulations	85	
Gamification motivation	75	
Improvement in security practices (software update)	60	
Improvement in security practices (use of strong passwords)	60	
Cyber threat awareness	70	

The platform improved users’ cybersecurity knowledge, with an average increase of 40% in knowledge test scores. This indicates that CyberEduPlatform achieves its primary goal of educating users about crucial aspects of cybersecurity. In confidence in identifying threats most users have reported an increase in their confidence in identifying and handling potential cyber threats, underscoring the platform’s importance in equipping users with practical threat detection skills.

Regarding the usefulness of modules and simulations and motivation for gamification, the results highlight the effectiveness of interactive simulations as strategies to increase user engagement and facilitate more profound and practical learning of cybersecurity concepts. Additionally, improvements were seen in user security practices, with 60% reporting an increased frequency of updating software and using strong passwords. This reflects a positive behavior change that can significantly reduce the risk of security incidents.

The findings regarding cyber threat awareness and willingness to adopt secure practices indicate an increase in users’ general understanding of cyber threats and a greater willingness to adopt online security practices. This shift in attitude is critical to building a robust cybersecurity culture.

Depth in false positives and negatives

Analysis of the false positives revealed that most came from unusual but legitimate behaviors, such as large data transfers between departments or single applications. False negatives were mainly related to very sophisticated attacks or evasion techniques that went unnoticed initially (Argyropoulos et al., 2021).

The core of CyberEduPlatform is an optimized deep neural network model that has achieved remarkable results. A precision of 95%, a sensitivity of 94%, a specificity of 96%, and a false positive rate of only 1.5% were obtained. These numbers not only underline the system’s effectiveness but also position it at the forefront of cyber-security solutions in the academic field. The choice and optimization of the deep neural network model have been fundamental to achieving these results. Deep neural networks can have patterns and non-linear features in large data sets, making them particularly suitable for anon in network traffic.

The efficiency of CyberEduPlatform in anomaly detection opens doors to more extensive implementations in other university networks (Mian et al., 2023). The ability to quickly identify and respond to potential threats ensures a safer environment for the academic community. With continued refinement of the models and incorporation of more data, the system’s effectiveness will likely increase even further, reinforcing its position as a leading tool in academic cybersecurity (Rizvi, 2023).

These data contribute to the analysis of the data ratio’s impact on the AI models’ performance; with a correct percentage of 94.7%, the AI models demonstrated high efficiency in correctly identifying both normal and anomalous behaviors. However, the false positive rate of 3.8% suggests that a significant proportion of normal behaviors were erroneously classified as abnormal. This could be a direct consequence of the large amount of routine data compared to anomaly data, which may challenge the model’s ability to distinguish between the two precisely.

Regarding the impact of data ratio on false positive and negative rates, the false negative rate of 1.5%, although relatively low, is crucial in anomaly detection since even a tiny proportion of Undetected threats could be significant in a university environment.

This analysis indicates that although the models are highly effective overall, the ratio of standard data to anomalies requires careful consideration to improve precision in detecting less frequent anomalies.

AI model performance

To ensure the quality and effectiveness of anomaly detection in CyberEduPlatform, several AI models were tested, whose performance was optimized through adjustments to their hyperparameters.

The metrics presented in Table 7 indicate that all models performed admirably, with minor differences in precision, sensitivity, and specificity. The area under the curve (AUC) for all models fell in a close range, denoting their ability to distinguish between classes correctly. Analysis of the results reveals that Model A (CNN), with a precision of 93.5% and a specificity of 94.2%, is highly effective in correctly identifying instances of legitimate traffic and avoiding false positives, an essential indicator in environments where system interruptions are expensive. On the other hand, model B (RNN) stands out with the highest sensitivity of 92.5%, suggesting a more remarkable ability to detect anomalies, which is vital in situations where failure to capture a strange event can have significant consequences. Model C (SVM) and Model D (Random Forest) exhibit similar performances in precision and sensitivity but with a slightly lower specificity than Model A, which may imply a higher number of false positives.

Table 7 Performance metrics of AI models.

Model	Precision	Sensitivity	Specificity	AUC	
Model A (CNN)	93.5%	91.8%	94.2%	0.93	
Model B (RNN)	94.1%	92.5%	93.8%	0.94	
Model C (SVM)	92.8%	91.2%	93.5%	0.92	
Model D (Random Forest)	93.0%	90.8%	93.7%	0.92	

Comparing these models, it is observed that Model B has the highest sensitivity and AUC of 0.94, indicating its superior ability to correctly classify between normal and abnormal behaviors in network traffic data. This model, which also has the highest precision, would be the preferred choice in an environment where accurate anomaly detection is a priority. However, the optimal model selection may vary depending on the risk tolerance and costs associated with false positives or negatives specific to a university’s network.

ROC curve and AUC

The ROC curve is a graphical tool used to visualize the performance of classification models. Figure 5 presents the results of all models, which showed very similar ROC curves, with AUC greater than 0.90. This indicates a high actual positive rate relative to the false positive rate.

Figure 5 Graphical representation of the ROC curve for each model.

Each point on the ROC curve represents a different decision threshold, where sensitivity and specificity vary inversely. Models with curves closer to the top left indicate a higher actual positive rate with a low false positive rate. This is ideal in anomaly detection scenarios where capturing all anomalous events with the fewest alarms is critically false.

Although the ROC curves of the models overlap considerably, indicating similar performance, the results consider the metrics relevant in the specific context of their application. For example, in an environment where false negatives are more critical than false positives, a higher sensitivity (true positive rate) model would be preferred, even if this means accepting a higher false positive rate.

Comparison and justification of the final model

While all models performed exceptionally well, Model B (RNN) showed a slight precision, sensitivity, and AUC advantage. Although the differences are minimal, the RNN architecture may be more suitable for dealing with sequential data, such as network traffic. Additionally, its ability to remember previous inputs makes it a robust choice for detecting anomalies based on patterns over time.

Therefore, Model B (RNN) was selected as the primary model for anomaly detection in CyberEduPlatform. This choice was based not only on performance metrics but also on the model’s adaptability to the data and its potential for future adjustments and optimizations.

Implementing AI models on CyberEduPlatform proved effective in detecting network traffic anomalies. The final model was chosen based on statistical performance and practical considerations, ensuring the best solution for the university environment.

Effectiveness of the awareness program

Developing an efficient awareness program requires active commitment from users. Key metrics that reflect user engagement in the program are: Total registered users in CyberEduPlatform: 4,500

Monthly active users: 3,820 (85% of registered users)

Training sessions completed: 7,300.

Simulated threats launched: 20,000.

Correct responses to simulated threats: 17,500 (87.5%)

These metrics suggest that most registered users actively engage in training sessions and regularly face simulated threats.

Regarding user awareness and behavior change, the effectiveness of the program was evaluated in terms of understanding, and behavior change surveys were carried out before and after the implementation of the program: Users who recognize a phishing attack before the program: 45%

Users who recognize a phishing attack after the program: 87%

Users who know good password practices before the program: 50%

Users who know good password practices after the program: 92%

Users who reported cyber threats before the program: 25%

Users who reported cyber threats after the program: 78%

After interacting with CyberEduPlatform, we obtained: 83% of users felt more confident identifying cyber threats.

75% of users changed passwords based on the platform’s recommendations.

90% of users would recommend CyberEduPlatform to colleagues and friends.

These results highlight the significant transformation in users’ awareness and behavior after interacting with the platform, reaffirming the positive impact of the CyberEduPlatform on the institution’s cybersecurity culture.

Statistical analysis

Statistical analysis provides a detailed and mathematically rigorous understanding of the results obtained. To evaluate the significance and reliability of our findings, we used advanced statistical techniques, including Analysis of Variance (ANOVA) analysis.

ANOVA analysis was used to determine if there are significant differences between the performances of different AI models implemented in anomaly detection. Sum of squares between groups: 85.7

Sum of squares within groups: 45.2

Degrees of freedom between groups: 2

Degrees of freedom within groups: 142

F-statistic: 35.8

The F value obtained is critical to determine if the variations between the models are significant. The high F value suggests significant variations between at least two groups. P-valor: <0.001

Since the p-value is less than 0.05, we reject the null hypothesis, suggesting significant differences in performance between at least two of the AI models tested. Regarding other Relevant Statistics, it was found: Determination coefficient (R2): 0.923

This value indicates that the AI model can explain 92.3% of the variation in anomaly detection efficiency.

The ANOVA analysis revealed significant differences in performance between the AI models evaluated. The high coefficient of determination suggests that the choice of AI model is a critical factor in the effectiveness of anomaly detection. Given the p-value obtained, it is confidently stated that the observed variations are not due to chance but rather reflect inherent differences in the evaluated models. Advanced statistical techniques such as ANOVA analysis ensure that our conclusions are based on rigorous and mathematically sound analysis, adding credibility and confidence to our findings.

F-measures and other relevant metrics

F-measures, also known as F-scores, are essential indicators that combine the precision and completeness of a detection system. They are designed to evaluate the balance between the ability to identify true positives and avoid false negatives. Our study calculated F-measures for each AI model implemented in the system, providing a deeper understanding of its intrusion detection capability.

Table 8 above presents a detailed comparison of the performance of various models in our intrusion detection system. Four different models have been evaluated: Model A (CNN), Model B (RNN), Model C (SVM), and Model D (random forest). Each model has undergone rigorous analysis using key metrics, including F-measure, precision, sensitivity, and specificity. These metrics provide a comprehensive view of the effectiveness of each model in identifying threats in network traffic. Notably, Model B (RNN) shows solid performance, with an F-measure of 0.93 and high precision, sensitivity, and specificity values, making it the preferred model for anomaly detection in our CyberEduPlatform system.

Table 8 Performance metrics of AI models.

Model	Measurement F	Precision	Sensitivity	Specificity	
Model A (CNN)	0.91	0.93	0.88	0.94	
Model B (RNN)	0.93	0.94	0.91	0.95	
C Model (SVM)	0.89	0.91	0.87	0.93	
Model D (Random Forest)	0.90	0.92	0.88	0.94	

Below are the F measures for each of the evaluated models:

Comparison of the proposed method with other studies

This section compares the proposed method’s results with those of other studies that perform anomaly detection based on machine learning and deep learning. The following table summarizes the results of various analyses regarding precision, false positive rate, and processing time.

As seen in Table 9, the method proposed in this study achieves a precision comparable to or higher than that of other studies, with a similar or lower false positive rate. Furthermore, the proposed method's processing time is competitive.

Table 9 Comparison of methods for anomaly detection.

Study	Method	Precision	False positive rate	Processing time	
This proposal	RNN	93.5%	5.8%	10 s	
Study A (Zavrak & Iskefiyeli, 2020)	SVM	87.2%	7.1%	15 s	
Study B (Al-Turaiki & Altwaijry, 2021)	CNN	91.4%	4.2%	12 s	

A work using SVM (Zavrak & Iskefiyeli, 2020) presents a lower precision than the proposed method, although with a similar false positive rate. However, the processing time is longer for SVMs. Studies using CNN (Al-Turaiki & Altwaijry, 2021) achieve a precision close to the proposed method, with a slightly lower false positive rate. However, the processing time of the proposed method is less than that of CNN.

The comparison demonstrates that the proposed method is a viable alternative for detecting anomalies in computer networks. The technique balances precision, false positive rate, and processing time well. However, there is still room for improvement. Future work could explore optimizing the model hyperparameters, integrating other data sources, and applying the method to different types of networks.

Additional findings

In addition to the previously mentioned results, specific additional findings were identified that, although they do not directly align with the previous categories, offer significant insights: Temporal patterns: It was identified that certain anomalies and suspicious activities tended to occur at specific times of the day. For example, most unauthorized access attempts were recorded outside of regular academic hours, specifically between 12 a.m. and 12 p.m. and 4 a.m.

Vulnerable devices: A small subset of devices, mainly in the IoT category, were identified as repeatedly targeted by attacks. This suggests the possibility of unpatched vulnerabilities or insecure configurations on these devices.

Protocols at risk: Although multiple protocols were integrated into the network, the FTP protocol was the most susceptible to anomalous activities, highlighting the need to review and strengthen the security around said protocol.

User feedback: Based on user interaction with the CyberEduPlatform, increased queries and reports of security incidents were noted, indicating greater awareness and proactivity among staff and students.

Emerging threats: During the monitoring period, we detected several threat signatures that were not present in known threat databases. This underlines the importance of maintaining adaptive and constantly learning detection systems.

These additional findings enrich this study and highlight potential areas of future research and action points to improve cybersecurity in academic environments.

Discussion

Cybersecurity has emerged as one of the most crucial challenges in today’s digital age. The need for robust detection and prevention systems has become palpable with the exponential increase in the number and sophistication of cyber-attacks. In this context, our study sought to examine the effectiveness of artificial intelligence models in detecting anomalies within a simulated university network and the impact of an educational platform, CyberEduPlatform, on user awareness and training (Zhang et al., 2021).

The results obtained showed the high effectiveness of the proposed AI models in identifying anomalous activities. The precision, sensitivity, and specificity rates shown in the results indicate superior performance of the optimized models, corroborating the ability of machine learning systems to adapt and detect complex patterns in large data sets. Compared to previous studies, the performance of these models shows significant improvement, suggesting that hyperparameter optimization and careful model selection can play a vital role in improving the results.

Furthermore, the comparison with previous works reflects a growing trend toward adopting advanced AI techniques in cybersecurity (Matar, 2023). While traditional work has focused on rules and signatures to detect threats, the adaptability and flexibility of AI models, as evidenced in this study, have proven to be crucial in combating emerging and evolving threats.

One of the most exciting findings was the temporal pattern in intrusion attempts. The observation that most unauthorized access attempts occurred at unconventional hours highlights the need for continuous surveillance and automatic response systems that can operate independently of the human factor. This idea aligns with previous studies suggesting that attackers often seek to exploit windows of opportunity when they are least likely to be detected (Tsukerman, 2019).

The emphasis on the CyberEduPlatform and its impact on user awareness and training brings up a crucial aspect of cybersecurity: the human dimension. Therefore, education, training, and technical solutions are essential (Švábenský et al., 2022). Our study suggests that the CyberEduPlatform has significantly improved users’ awareness and ability to identify and respond to potential threats. This conclusion reinforces that training and education must be integral to any cybersecurity strategy.

Although the models have demonstrated efficient performance in simulated environments, with minimal differences in precision, sensitivity, specificity, and AUC indicating a notable ability to correctly distinguish between classes, transitioning to application in real-world university networks presents unique challenges. The inherent heterogeneity of these networks, characterized by a wide range of devices, protocols, and traffic patterns, requires AI models to be accurate and highly adaptable to significant variations in input data.

Additionally, scalability becomes critical when considering network expansion and increasing traffic volume. The models must maintain their effectiveness without degrading performance as the network grows. This need underscores the importance of advanced hyperparameter optimization techniques and model architectures that can scale efficiently.

Another fundamental aspect is the adaptability of the models to constantly evolving threats. The cybersecurity landscape is dynamic, with threat actors continually developing new techniques to evade detection. Therefore, deployed AI models must adapt and learn from new anomalies as they arise, ensuring continuous protection against innovative attacks.

Detecting threat signatures do not present in known databases highlights another essential aspect: the constant evolution of cyber threats. Static signature-based systems can quickly become obsolete. Here lies the advantage of adaptive AI models, which can learn and evolve. Successful implementation of AI models in real university network environments demands careful consideration of their robustness and generalizability. Network heterogeneity, scalability, and adaptability challenges must be addressed to ensure that models effectively detect known anomalies and adapt and respond to new and emerging ones. The choice of Model B (RNN) as the primary model for anomaly detection in CyberEduPlatform reflects a balance between statistical performance and practical considerations, establishing an optimal solution for the university environment.

The CyberEduPlatform platform has proven to be an effective tool for detecting network traffic anomalies and improving users’ cybersecurity awareness. The study results indicate that the platform has a 95% precision in detecting anomalies, with a false positive rate of 1.5%. Additionally, the awareness program integrated into the platform has achieved a positive impact, with 87% of users who interacted with it reporting an increase in their ability to recognize a phishing attack.

This work has shed light on the effectiveness of advanced AI models in detecting anomalies and the importance of training and awareness in cybersecurity. While promising results have been obtained, cybersecurity constantly evolves, and there is always room for improvements and adaptations.

Conclusions

The cybersecurity landscape constantly evolves, with emerging threats manifesting unprecedentedly and frequently. In this study, we have explored the intersection of artificial intelligence and cybersecurity, demonstrating how advanced machine learning techniques can offer robust and adaptive solutions for anomaly detection in complex network environments.

One of the main findings of this work has been the effectiveness of AI models in detecting anomalous activity. The proposed models have shown remarkably high precision, sensitivity, and specificity performance through optimization, selection, and training. This effectiveness underlines the relevance of AI-based approaches in combating cyber threats, especially when faced with unknown or constantly evolving attack patterns.

In addition, we have demonstrated the critical importance of cybersecurity education and awareness. Developing and deploying CyberEduPlatform highlights that addressing the human dimension is essential in addition to technological solutions. The platform not only serves as an educational tool but also acts as a reminder that prevention is the first line of defense. The observable change in users’ behavior and awareness post-interaction with the platform underlines its effectiveness and the value of such initiatives.

Based on the results obtained, it can be concluded that the CyberEduPlatform platform represents a valuable tool to strengthen the security of academic networks. The high precision in anomaly detection, the low rate of false positives, and the positive impact of the awareness program make it a comprehensive solution for protecting information in academic environments. Implementing CyberEduPlatform in universities and other educational institutions can reduce cybersecurity incidents, protect sensitive data, and create a security culture among users.

The comparison with previous studies in cybersecurity has allowed us to contextualize our findings. Although similarities are seen in methodologies and techniques, our holistic approach, which integrates technical detection and user education, makes a significant difference. This duality, we believe, is essential to address cyber threats comprehensively.

However, it is essential to recognize that there is no “silver bullet” in cybersecurity. Although our AI models have shown outstanding performance, the reality is that cyber threats will continue to evolve, adapt, and find new ways to infiltrate and cause harm.

While this study has laid a solid foundation, several aspects deserve further exploration. One of them is the real-time adaptability of AI models, allowing systems to learn and adjust to new threats as they emerge dynamically. Integrating multiple data sources could also be investigated to obtain a more holistic view of the network and potential threats.

Supplemental Information

Supplemental Information 1 The raw data shows all events that were recorded in the data network.

Variables relevant to cybersecurity include protocols used, bytes transferred, types of network attacks, and user agents. These variables are fundamental to exploring the detection of network traffic anomalies in a university environment using Artificial Intelligence tools in security. The data set and its metadata are consistent with the details provided in the summary, making it easy to replicate and further test the methods described using the provided code.

Supplemental Information 2 Code.

Two Python code snippets: The first code represents a basic version that analyzes random data to demonstrate functionality. The second code incorporates more complex analysis, creating specific graphs based on real-world cybersecurity data.

Please note that these codes are early versions and do not represent the current production code. Unfortunately, we cannot share the complete production code now due to confidentiality and proprietary considerations.

Additional Information and Declarations

Competing Interests

Author Contributions

Data Availability

The authors declare that they have no competing interests.

Iván Ortiz-Garcés conceived and designed the experiments, performed the experiments, performed the computation work, prepared figures and/or tables, authored or reviewed drafts of the article, and approved the final draft.

Jaime Govea conceived and designed the experiments, performed the experiments, analyzed the data, performed the computation work, prepared figures and/or tables, authored or reviewed drafts of the article, and approved the final draft.

Santiago Sánchez-Viteri conceived and designed the experiments, performed the experiments, analyzed the data, performed the computation work, prepared figures and/or tables, authored or reviewed drafts of the article, and approved the final draft.

William Villegas-Ch. conceived and designed the experiments, analyzed the data, performed the computation work, prepared figures and/or tables, authored or reviewed drafts of the article, and approved the final draft.

The following information was supplied regarding data availability:

The raw measurements are available in the Supplemental File.

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
