# Peer review of "CyberEduPlatform: an educational tool to improve cybersecurity through anomaly detection with Artificial Intelligence"

_PeerJ Computer Science, doi:10.7717/peerj-cs.2041_

## Round 0.1 · original submission · Major Revisions

The review process is now complete. While finding your paper interesting and worthy of publication, the referees and I feel that more work could be done before the paper is published. My decision is therefore to provisionally accept your paper subject to major revisions.

**Language Note:** PeerJ staff have identified that the English language needs to be improved. When you prepare your next revision, please either (i) have a colleague who is proficient in English and familiar with the subject matter review your manuscript, or (ii) contact a professional editing service to review your manuscript. PeerJ can provide language editing services - you can contact us at [email protected] for pricing (be sure to provide your manuscript number and title). – PeerJ Staff

Reviewer 1 ·

Basic reporting

Your research on cybersecurity and artificial intelligence demonstrates a comprehensive understanding of the subject matter.

Methodological Clarity: Provide more detailed information about the methodology employed in evaluating AI models for anomaly detection. Specify the criteria used for model selection, training, and evaluation. Additionally, elaborate on the simulation setup, including parameters and configurations, to facilitate reproducibility and validate your findings.

Experimental design

Data Preprocessing and Feature Engineering: Describe the preprocessing steps and feature engineering techniques applied to the network traffic data. Clear documentation of these processes is essential for understanding how the input data were transformed and prepared for AI model training. Consider discussing any challenges encountered during data preprocessing and how they were addressed.

Model Performance Evaluation: Provide a comprehensive analysis of the AI models' performance metrics, including accuracy, precision, recall, and F1 score. Present comparative results to highlight the strengths and weaknesses of each model in detecting anomalies within network traffic. Consider using visualizations such as ROC curves or confusion matrices to elucidate the performance differences effectively.

Robustness and Generalization: Assess the robustness and generalization capabilities of the AI models beyond the simulated environment. Discuss potential challenges and limitations in deploying these models in real-world university network environments, considering factors such as network heterogeneity, scalability, and adaptability to evolving threats.

Validity of the findings

Provide more details about the design and implementation of the CyberEduPlatform for raising cyber awareness. Describe the platform's features, content, and user engagement strategies. Consider conducting user studies or surveys to evaluate the platform's effectiveness in improving cybersecurity awareness among its users.

Reviewer 2 ·

Basic reporting

The paper proposes a anomaly detection system supported by machine learning and deep learning, along with a real-time monitoring platform for awareness and education purposes.

- The contribution has not been emphasized well. It should be provided in a clearer and more detailed manner.
- According to the journal format, references should all be presented in the same style.
- The literature review is inadequate. There are many studies on anomaly detection based on machine learning and deep learning. These should be mentioned in the related works section, and their results should be compared in the "Discussion" section.
- Information resembling preliminaries is provided in all sections. A section and figure detailing the general structure and specifics of the proposed method can be added.
- “This data included logs of IP addresses, source and destination ports, protocols used, and traffic volumes.”
Based on this sentence, what are the features in the proposed deep and machine learning models? Each one should be explained.
- How are the output data labeled when creating the dataset? In other words, if there are two labels (there is no detail in the paper about this), how are the cases where attributes are labeled as anomalies or normal determined? It should be explained.
- Numeric results obtained should be provided in the Abstract and Conclusion sections.

Experimental design

- Which preprocessing methods (eliminating data missing, normalization, etc.) were used when preparing the dataset? This should be explained in detail.
- “The models considered included convolutional neural networks (CNN), recurrent neural networks (RNN), support vector machines (SVM), and random forests (Random Forest).”
The reasons for choosing these methods should be explained.
- The paper describes two cases: the network monitoring module and the processing of data obtained through this module for anomaly detection. However, there is no information about the network monitoring tool itself. It is only mentioned that terabytes of data were obtained using such a system. Detailed information is needed about the infrastructure and methods used in the network monitoring model.

Validity of the findings

- The F-score, similar to precision, sensitivity, specificity, and AUC, should be explained and formulated.
- In the "Preventing Phishing Attacks" section, it is mentioned that the system detects and prevents phishing attacks. It is stated that anomaly detection is done through classification. However, how the prevention process is carried out is not clear. Prevention procedures should be explained and analyzed.
- The results of the proposed method should be compared with the results of other studies that perform anomaly detection based on machine learning and deep learning.

---

## Round 0.2 · Major Revisions

The reviewer pointed out some problems with the manuscript. Thus, my decision is major revision.

Reviewer 1 ·

Basic reporting

Authors updated the paper.

Experimental design

As above

Validity of the findings

As above

Reviewer 2 ·

Basic reporting

- Figure 1 illustrates the steps of a general machine learning, but there should be a customized figure according to the proposed model that enables readers to understand your study through this figure. For example, there are no details about data collection, and the caption of the figure is "Diagram of Anomaly Detection Methodology in University Network Traffic," but there is no indication of security in the figure. Therefore, a more detailed drawing specific to the addressed problem is required.

Experimental design
* * *
Validity of the findings

- The formulas of the metrics are provided in the "Proposed Methodology" section. However, the formula for the f-measure is not included. This formula should also be added. How was the f-measure calculated? It should be stated.

- In the section titled "Preventing phishing attacks," general information about the topic is provided. Detecting and preventing attacks are different concepts. You mentioned applying machine learning and deep learning for detection in your proposed method, but was anything specifically done for prevention? Do you have any contribution in terms of prevention? Why is such a section included?

- References to the articles in Study A and B in Table 9 should be made, and comparisons should be transparent and verifiable. With which studies in the literature did you compare your method?

---

## Round 0.3 · accepted · Accept

Since the reviewers' comments have been addressed, we are happy to inform you that your manuscript has been accepted for publication.

Reviewer 2 ·

Basic reporting

The authors have made the necessary revisions.

Experimental design

The authors have made the necessary revisions.

Validity of the findings

The authors have made the necessary revisions.